# Prevalence and covariates of depression among older adults in Nepal: A systematic review and meta-analysis

**Gayatri Khanal**[1]*, **Y. Selvamani**[1], **Suman Thapa**[2], **Saravanan Chinnaiyan**[1]

**1** SRM Institute of Science and Technology, Chennai, Tamilnadu, India, **2** Nisarga Hospital and Research Centre, Dhangadi, Kailali, Nepal

* khanalgayatri2@gmail.com, gk3837@srmist.edu.in

**Data Availability Statement:** This study incorporates all of the data created or examined in the review.

## Abstract

Globally, depression is the most common mental health condition in older individuals which is a significant emerging public health problem in developing countries like Nepal. Older adults with depression are overlooked in Nepal due to the paucity of updated evidences on geriatric depression. The current study is aimed to determine the prevalence of depression and its covariates among older adults. PRISMA-compliant searches of the PubMed, Scopus, NepMed/Nepjol, and Google Scholar databases were conducted from 2013 to 2023. The included papers' quality was assessed using a JBI quality appraisal technique. The data were analysed using R statistical program. Heterogenicity was assessed by $I^2$ test. Random-effects model was used to calculate the pooled prevalence of depression. Twenty studies from three regions of Nepal including 5728 older adults were assessed. In Nepal, geriatric depression affected 52% of the overall population (95% CI = 44%, 59%). The pooled prevalence in the subgroup analysis was higher in the central region and among older adults living in old age homes. The presence of chronic diseases, $\geq 70$ years of age, female, illiterate, limitation of Instrumental activities of daily living (IADL), and feeling of loneliness were the independent predictors of geriatric depression. Egger (p < 0.0002) and Harbord (p < 0.0001) indicated the presence of publication bias. Even though the estimation of depression differs with geographic region and study settings, one out of two older adults in Nepal had depression, suggesting the need for public health interventions to support and reduce geriatric depression in Nepal.

## Introduction

Geriatric depression is a mental health condition which affects older people and characterized by a low mood or loss of pleasure for extended period of time usually more than two weeks that interfere with daily activities and quality of life [1]. Detecting depression early in older adults can be difficult because it often presents differently, with physical symptoms such as body aches and decreased cognitive function [1, 2]. Symptoms can vary and include feeling of numbness, fatigue, and lack of concentration [2]. Additionally, older adults face common risk

**Funding:** The authors received no specific funding for this work.

**Competing interests:** The authors have declared that no competing interest exist.

factors like chronic illnesses, reduced functional abilities, social isolation, and loneliness more frequently than younger people [2, 3]. Moreover, the symptoms of geriatric depression are similar with the aging process like feeling tiredness, difficulty in concentration, irritability, change of appetite and weight, confusion, feeling hopeless, worthless, or guilty [2, 4]. Consequently, during the initial stage of depression, these common symptoms in older adults are frequently ignored which has a devastating consequence contributing significantly to misery in late life [3].

Depression can be easily prevented among older people through early identification process and proper management. However, it has resulted in significant mortality, morbidity, and a decline in physical, cognitive, and social functioning as well as in overall life quality [5]. In addition, depression has a substantial impact on the prognosis of chronic illnesses, which worsens impairment. As a result, the suicidal mortality and non-suicidal morbidity rates are much greater in elderly people with depression [6]. These demands unveiling of the current prevalence of depression, particularly in the low- and middle-income countries like Nepal.

With an increase in the older population growth rate of 3.29% per year, which is about 3.5 times greater than the national population growth rate (0.92%), the proportion of the older population in Nepal has grown rapidly over the past few decades [7, 8]. Based on the 2021 census, the population of older adults in Nepal has increased by 38.2% compared to the data obtained from the 2011 census, resulting in a total of 2.97 million individuals. At present, approximately 10.2% of the population in Nepal consists of older age individuals [8].

The increasing share of the older population experiences several social and health challenges, including elevated risk of chronic diseases, disability from illness, lack of family support, bereavement, limited personal autonomy, loneliness, and financial dependency, which can negatively impact mental health, particularly depression [4, 5, 9, 10]. The most prevalent mental health illness, with significant global cause of disability and burden, is depression [5, 11]. Similarly, in older people, the common psychiatric illness is depression, which is a significant predictor of mortality and disability [11–13]. WHO report identified that depressive disorders affect over 300 million (10 to 20%) of elderly populations worldwide [14]. Nonetheless, recent systematic review revealed that prevalence of depressive symptoms among the elderly in poor countries was higher (40.78%) than rich countries (17.05%) [15].

Studies conducted in Nepal estimate the prevalence of geriatric depression range from 15.4% to 82.6% [16, 17]. These variations differ with settings, gender, race and other associated factors. Nepal is a low-middle-income country where the health care system is evolving and has not yet prioritized mental health care including depressive disorders among older population [18]. Depression is one of the most common problems among the older people of Nepal where people have difficulty in controlling their mood leading to an adverse impact on their daily life [17, 18]. There are various risk factors responsible for depression in older adults which are females, widowed, separated/ or divorced [19, 20], living alone [21], lack of formal education [21], presence of physical diseases [21], feeling of loneliness [22], poor economic status [23, 24], functional impairment [25] and poor social support [20]. Despite the scenario, there is paucity of the nationally representative evidences on geriatric depression.

Even though there are several small-scale studies conducted revealing inconsistent prevalence on elderly depression in Nepal, meta-analyses of the epidemiological studies have not yet been conducted thoroughly using reliable screening tools. In this perspective, we need current epidemiological evidence of geriatric depression which will be useful for the public health policy in Nepal for proper planning of geriatric depression diagnosis, treatment and management. Furthermore, this study may help to formulate suggestions for clinicians, the research community, and policy makers. Hence, the present study was conducted with the aim of estimating the prevalence of depression among older population and citing its contributing factors.

## Methodology

### Reporting

The current systematic review and meta-analysis was conducted to report the findings and the methodology used to choose the included studies (S1 File) was a according to the Preferred Reporting Items of Systematic Reviews and Meta-Analysis (PRISMA) checklist [26].

### Study protocol registration

The International Prospective Register of Systematic Reviews (PROSPERO) has recorded the protocol of this review under registration number CRD42023454911/2023.

### Inclusion criteria

The PICOS acronym created for this study served as the basis for the inclusion criteria, which are as follows: Participants (P): Senior citizens 60 years of age and older; exposure or intervention: not applicable; No comparisons should be made; Outcomes include the prevalence of depression as measured by any reliable method and cross-sectional studies as the research design (S). The studies published in English between January 2013 and August 2023 were chosen.

### Exclusion criteria

Geriatric depression found in particular communities was not included (such as ethnic groups like the Rai). Additionally disregarded were studies done during the COVID-19 epidemic. The study excluded review articles, editorials, conference abstracts, commentaries, and pieces without the full text.

### Literature source and search strategy

To identify the pertinent publication published between these databases, several electronic databases such as PubMed, Scopus, NepJol, NepMed, and Google Scholar were examined. The search strategy was built by a combination of subject terms, text terms and the Medical Subject Headings (MESH) terms for instance "depression", "geriatric depression", "depressive symptoms" "elderly", "old age", "senior citizen", and Nepal and Boolean operators like "AND" and "OR". The complete information on the search strategy of this study is reported in S2 File.

### Study selection and data extraction

Two separate authors (GK and SC) performed a first-stage screening of the titles and abstracts of all the retrieved articles based on the inclusion criteria. The full texts of the qualified papers were further examined in compliance with the inclusion criteria. Selected final studies were chosen for this trajectory. To extract data from individual research, pre-designed, standardized data extraction forms have been created based on the Cochrane Good Practice Data Extraction Template [27]. For the purposes of study selection, quality evaluation, and data extraction, the two authors (GK and SC) both worked independently. Any differences of opinion on a specific paper were resolved after contacting with other two authors (YS and ST).

### Statistical analysis

In order to establish pooled effects, subgroup analysis, publication bias analysis, forest plot, and sensitivity analysis, the meta-analysis was carried out using the R software, which was created by the R Foundation for Statistical Computing in Vienna, Austria. I squared statistics ($I^2$)

were used to calculate the heterogeneity between studies. The current study used the following criteria to quantify heterogeneity based on $I^2$ value. $I^2$ values of 25%, 50%, and 75%, which indicate low, moderate, and high heterogeneity, respectively, were within the range of 0% (No heterogeneity) and 100% (High heterogeneity). Between the studies, there was significant heterogeneity ($I^2$ = 98%). So, for analysis, we employed the random effects model [3]. Within a 95% confidence interval (CI), the outcomes obtained were presented as a polled prevalence of geriatric depression. To calculate the pooled prevalence, a forest plot was used. Using Egger's test and visual inspection of the asymmetry in funnel plots, the presence of publication bias was assessed [28–30]. P-values ≤ 0.10 were regarded as statistically significant for Egger's test. We conducted a subgroup analysis based on the geographical (Eastern, Western, and Central) and residential (Old Care Age Home or Community) setups of Nepal. This study also undertook sensitivity analysis to identify studies that had a significant influence on article's systematic review and meta-analysis.

### Risk of bias assessment/ quality appraisal

To evaluate the methodological quality of each study, the Joanna Briggs Institute (JBI) standard critical appraisal tool for prevalence studies was utilized [31]. JBI appraisal tool consists of nine items of questions. The studies' quality was evaluated independently by two researchers (GK and SC), and the other two (YS and ST) were consulted when there was a difference of opinions. A score of 0 was assigned for "No" or "unclear" or not applicable" response and 1 score for "Yes" response. The maximum possible quality score was 9, and the minimum was 0. Studies were categorized as poor (score = 0–4), moderate (score = 5–7), or high (score = 8–9) quality based on the score [32, 33]. Finally, studies with high or moderate quality were included (Table 1).

## Results

The search of PubMed identified 67 potentially relevant articles, Scopus recognized 89 articles, Google Scholar figured out 22 articles, and Nepjol and NepMed identified 7 articles. So, a total of 185 articles were retrieved. After removing 39 duplicate publications, 146 unique articles were chosen for review at the abstract level. Out of these, 43 articles were selected for further review with full-text. After the full-text review, only 20 articles met the study's inclusion criteria. The inclusion procedure was carried out in line with the PRISMA flowchart (Fig 1).

### Characteristics associated with the included studies

The summary of the included studies is mentioned in Table 2. In this study, 20 studies were included. Each included study had an average sample size of about 136. Cross-sectional design was used in all of the included studies. Overall, the included studies recruited 5728 participants. Eight of the 20 studies were carried out in old care age homes, while the remaining 12 were done in community settings. More than fifty percent (n = 12) of the published research articles were conducted in the Central, followed by Eastern (n = 4) and Western (n = 4) regions. Fourteen studies had applied the Geriatric Depression Scale-15 items scale, 4 used Geriatric Depression Scale -30 item scale and the remaining 2 studies utilized the International Classification of Disease-10 and Depression Anxiety Stress Scale as a diagnostic measurement tool for depression. The included studies were published between 2013 to 2023 [16, 17, 34–51].

### Prevalence of depression among older adults in Nepal

To estimate the prevalence of geriatric depression among the elderly in Nepal, we included data from 20 studies. The prevalence range among older persons was between 15.4% and

**Table 1. Critical appraisal of included studies by JBI critical appraisal tool.**

| Included studies | | Criterion No (Items included to appraise prevalence studies) | | | | | | | | | Overall quality | Decision |
|---|---|---|---|---|---|---|---|---|---|---|---|---|
| Author | Year | 1 | 2 | 3 | 4 | 5 | 6 | 7 | 8 | 9 | | |
| Rajan S et al., [34] | 2013 | Yes | No | No | Yes | No | Yes | Yes | Yes | NA | Moderate | Include |
| Chalise Hom Nath et al. [35] | 2014 | Yes | Yes | No | Yes | Yes | Yes | Yes | Un | NA | Moderate | Include |
| Kafle B et al., [36] | 2015 | Yes | Yes | Un | Yes | Un | Yes | Yes | Yes | Un | Moderate | Include |
| Gupta Ankit Kumar et al. [37] | 2016 | Yes | Un | Un | Yes | Un | Yes | Un | Yes | Yes | Moderate | Include |
| Sharma KR et al., [38] | 2018 | Yes | Un | Yes | Yes | Yes | Yes | Yes | Yes | NA | Moderate | Include |
| Ghimire Saruna et al., [39] | 2018 | Yes | Yes | Yes | Yes | Yes | Yes | Yes | Yes | NA | High | Include |
| Simkhada R et al., [40] | 2018 | Yes | Yes | Yes | Yes | Yes | Yes | Yes | Yes | Yes | High | Include |
| Devkota Rashmi et al., [41] | 2019 | Yes | Yes | Yes | Yes | Yes | Yes | Yes | Yes | NA | High | Include |
| Sapkota N et al., [42] | 2019 | Yes | No | No | Yes | Yes | Yes | Yes | Un | NA | Moderate | Include |
| Manandhar Kedar et al., [43] | 2019 | Yes | Yes | Yes | Yes | Yes | Yes | Yes | Yes | Yes | High | Include |
| Shrestha Kenison et al., [44] | 2020 | Yes | Un | NA | Yes | Un | Yes | Yes | Yes | Un | Moderate | Include |
| Yadav Uday Narayan et al., [45] | 2020 | Yes | Yes | Un | Yes | Yes | Yes | Yes | Yes | Yes | High | Include |
| Thapa Deependra K. et al., [16] | 2020 | Yes | Yes | Un | Yes | Un | Un | Yes | Yes | Un | Moderate | Include |
| Shrestha Roshana et al., [46] | 2020 | Yes | No | Yes | Yes | Yes | Yes | Yes | Yes | Yes | High | Include |
| Mali Prajita et al., [47] | 2021 | Yes | Yes | Yes | Yes | Un | Yes | Yes | Yes | Unclear | Moderate | Include |
| Sharma Muna et al., [48] | 2021 | Yes | Yes | Un | Yes | Yes | Yes | Yes | Yes | Yes | High | Include |
| Lamichhane Prava et al., [17] | 2022 | Yes | No | No | Yes | Yes | Yes | Un | Yes | Yes | Moderate | Include |
| Acharya Saurav Chandra Samadarshi et al., [49] | 2022 | Yes | Yes | Yes | Yes | Yes | Yes | Yes | Yes | NA | High | Include |
| Chapagain Sakuntala et al., [50] | 2022 | Yes | Yes | Yes | Yes | Un | Yes | Un | Yes | Un | Moderate | Include |
| Dhungana Ananta Raj et al., [51] | 2023 | Yes | No | No | Yes | No | Yes | Yes | Yes | Un | Moderate | Include |

NA = Not Applicable, Un = Unclear, yes = Criterion fulfilled, No/Unclear/Not applicable = Criterion fulfilled

Criterion No 1: Was the sample frame appropriate to address the target population? Items 2: Were study participants sampled appropriately? Items 3: Was the sample size adequate? Items 4: Were the study subjects and the settings described in detail? Item 5: Was the data analysis conducted with sufficient coverage of the identified sample? Item 6: Were valid methods used for the identification of the condition? Items 7: Were the condition measured in a standard, reliable way for all participants? Items 8: Was there an appropriate statistical analysis? Items 9: Was the response rate adequate?

82.6% [16, 17, 34–51]. In Nepal, among older individuals, the combined prevalence rate of depression was 52% (95% CI = 44%, 59%) (Fig 2). However, a significant amount of heterogeneity was found ($I^2$ = 98%, chi-square test for heterogeneity = $p < 0.001$). In eastern, western, and central Nepal, the estimated pooled prevalence of geriatric depression among the aged was 46% (95% CI, 25%-66%), 49% (95% CI, 19–79%), and 55% (95% CI, 48%-61%), respectively (Fig 3). Subgroup analysis was also performed between old age homes and communities centers. The calculated pooled prevalence of geriatric depression among older adults in old age homes was 58% (95% CI 45% to 71%) and 49% (95% CI 39% to 58%) in the community centers (Fig 4). To test the robustness of the results, a sensitivity analysis was done by removing one study at a time. According to the pooled prevalence estimations, the total prevalence was between 50% and 54% (95% CI, 43% to 58%) (Fig 5).

## Publication bias

In systematic reviews and meta-analyses, publication bias has been identified using two different methods. The funnel plot, as depicted in (Fig 6) did not exhibit substantial indications of publication bias in relation to the pooled prevalence rates. Nevertheless, the statistical tests conducted by Egger ($p < 0.0002$) and Harbord ($p < 0.0001$) indicated the presence of publication bias.

### Factors contributing for geriatric depression

Out of 20 included studies [16, 17, 34–51], only 10 studies [16, 35, 38, 40, 43, 46–48, 50] had commented on associated factors for geriatric depression. Narrative synthesis for the factors contributing for geriatric depression revealed that presence of chronic diseases [17, 40, 43, 47–50], age 70 or more [38, 43, 47, 49, 50], female sex [35, 38, 43, 48, 50], illiterate [40, 43, 50],single/window/divorced [38, 43, 50], unsatisfied [38, 43, 50], limitation of IADL [35, 40], feeling of loneliness [35, 40] were positively linked with geriatric depression (Table 3).

## Discussion

The goal of the current systematic review and meta-analysis was to provide updated national estimates on geriatric depression among older adults in Nepal. The evidences would be more

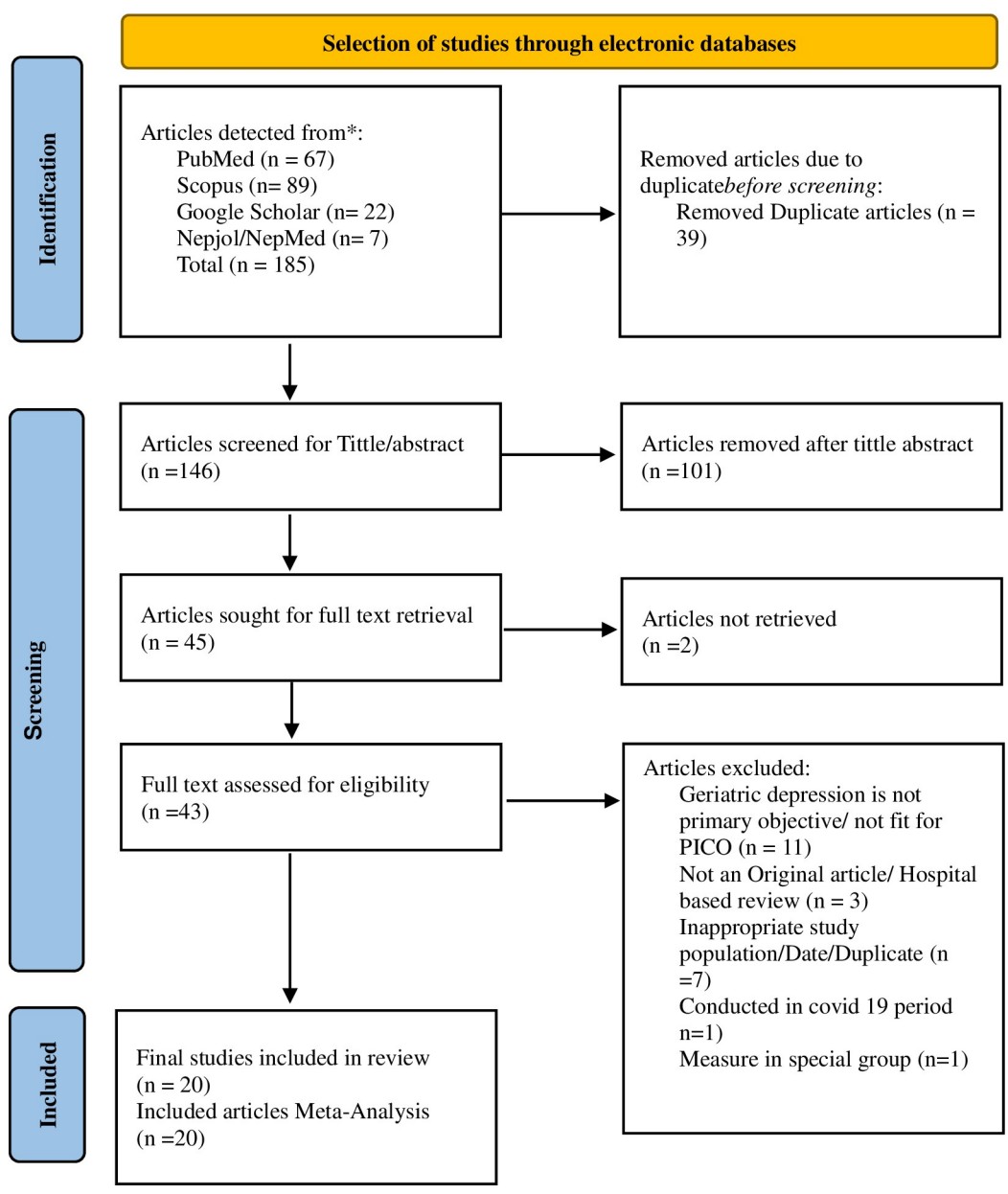

**Fig 1. Prisma flowchart for selection of the study.**

**Table 2. Characteristics of the studies included.**

| Study ID | Author name | Year of publication | Region | Study Settings | Study Design | Sampling Techniques | Sample size | Age range | Tool used | Total number of outcomes | Prevalence (%) | Do predictors reported |
|---|---|---|---|---|---|---|---|---|---|---|---|---|
| 1 | Rajan et al., [34] | 2013 | Central | Old age home | CSD | CE | 150 | ≥60 | GDS-30 | 71 | 47.33 | No |
| 2 | Chalise et al., [35] | 2014 | Central | Old age home | CSD | CE | 165 | ≥60 | GDS-15 | 104 | 57.8 | Yes |
| 3 | Kafle et al., [36] | 2015 | Central | Old age home | CSD | PS | 203 | ≥60 | GDS-15 | 96 | 47.3 | No |
| 4 | Gupta et al., [37] | 2016 | Eastern | Community | CSD | NPS | 189 | ≥60 | ICD 10 | 34 | 18 | No |
| 5 | Sharma et al., [38] | 2018 | Eastern | Community | CSD | PS | 353 | ≥60 | GDS-30 | 230 | 65.2 | Yes |
| 6 | Ghimire et al., [39] | 2018 | Central | Community | CSD | NPS | 289 | ≥60 | GDS-15 | 166 | 57 | No |
| 7 | Simkhada et al., [40] | 2018 | Central | Community | CSD | PS | 303 | ≥60 | GDS-15 | 175 | 60.6 | Yes |
| 8 | Devkota et al., [41] | 2019 | Central | Community | CSD | PS | 124 | ≥60 | GDS-15 | 63 | 50.8 | No |
| 9 | Sapkota et al., [42] | 2019 | Eastern | Old age homes | CSD | NPS | 62 | ≥60 | GDS-30 | 27 | 43.54 | No |
| 10 | Manandhar et al., [43] | 2019 | Central | Community | CSD | PS | 439 | ≥60 | GDS-15 | 246 | 56 | Yes |
| 11 | Shrestha et al., [44] | 2020 | Central | Old age home | CSD | NPS | 159 | ≥60 | GDS-15 | 63 | 39.6 | No |
| 12 | Yadav et al., [45] | 2020 | Eastern | Community | CSD | PS | 794 | ≥60 | GDS-15 | 443 | 55.8 | No |
| 13 | Thapa et al., [16] | 2020 | Western | Community | CSD | PS | 794 | ≥60 | DASS-21 | 122 | 15.4 | Yes |
| 14 | Shrestha R et al., [46] | 2020 | Central | Community | CSD | PS | 280 | ≥60 | GDS-15 | 128 | 45.7 | Yes |
| 15 | Mali et al., [47] | 2021 | Central | Old age home & community | CSD | PS | 244 | ≥60 | GDS-15 | 142 | 58.1 | Yes |
| 16 | Sharma et al., [48] | 2021 | Western | Community | CSD | PS | 245 | ≥60 | GDS-15 | 80 | 32.6 | Yes |
| 17 | Lamichhane et al., [17] | 2022 | Central | Old age home | CSD | NPS | 155 | ≥60 | GDS-15 | 128 | 82.6 | No |
| 18 | Samadarshi et al., [49] | 2022 | Western | Community | CSD | PS | 405 | ≥60 | GDS-15 | 279 | 68.9 | Yes |
| 19 | Chapagain et al., [50] | 2022 | Central | Community | CSD | PS | 318 | ≥60 | GDS-15 | 157 | 49.4 | Yes |
| 20 | Dhungana et al., [51] | 2023 | Western | Old age home | CSD | NPS | 57 | ≥60 | GDS-30 | 46 | 80.7 | No |

CSD = Cross-Sectional Study, PS = Probability Sampling, NPS = Non- Probability Sampling CE = Complete Enumeration

effective in strengthening public health policy regarding effective treatment and mental health promotion.

The World Health Organization estimation showed that depression among the older adults occurs globally with prevalence that typically ranges from 10% to 20% [14]. Nonetheless, the present study revealed that the magnitude of geriatric depression among the older population in Nepal was as high as 52%. This pooled estimate 52% (95% CI 44, 59) of geriatric depression among older adults in Nepal was higher than other systematic review and meta-analysis studies conducted across the globe by Zenebe et al., 2021, 31.74% (95% CI 27.90,35.59) [15], Wang

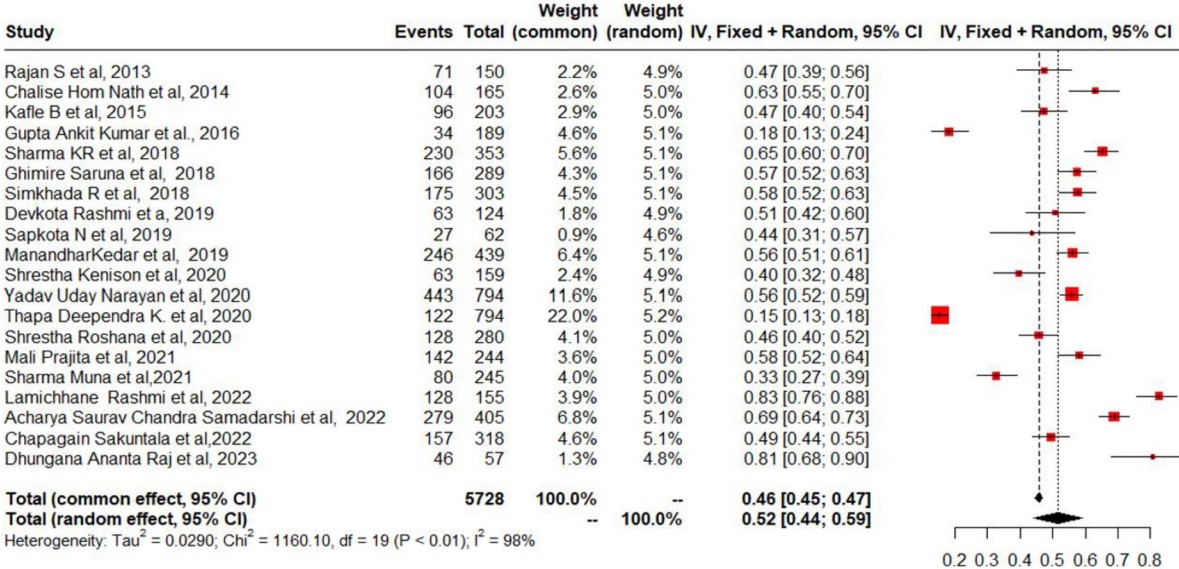

**Fig 2. Forest plot of combined frequency of geriatric depression among the elderly in Nepal.**

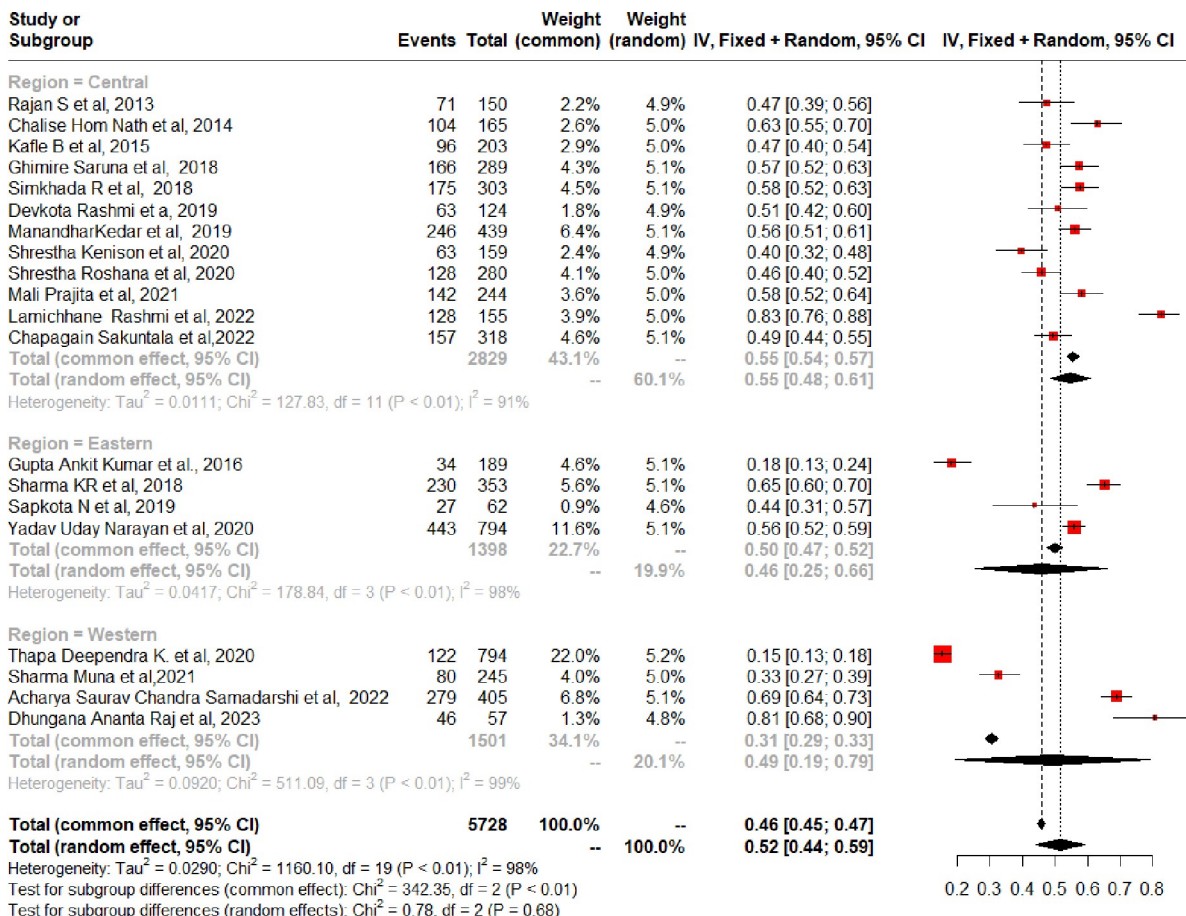

**Fig 3. Analysis of subgroups based on region.**

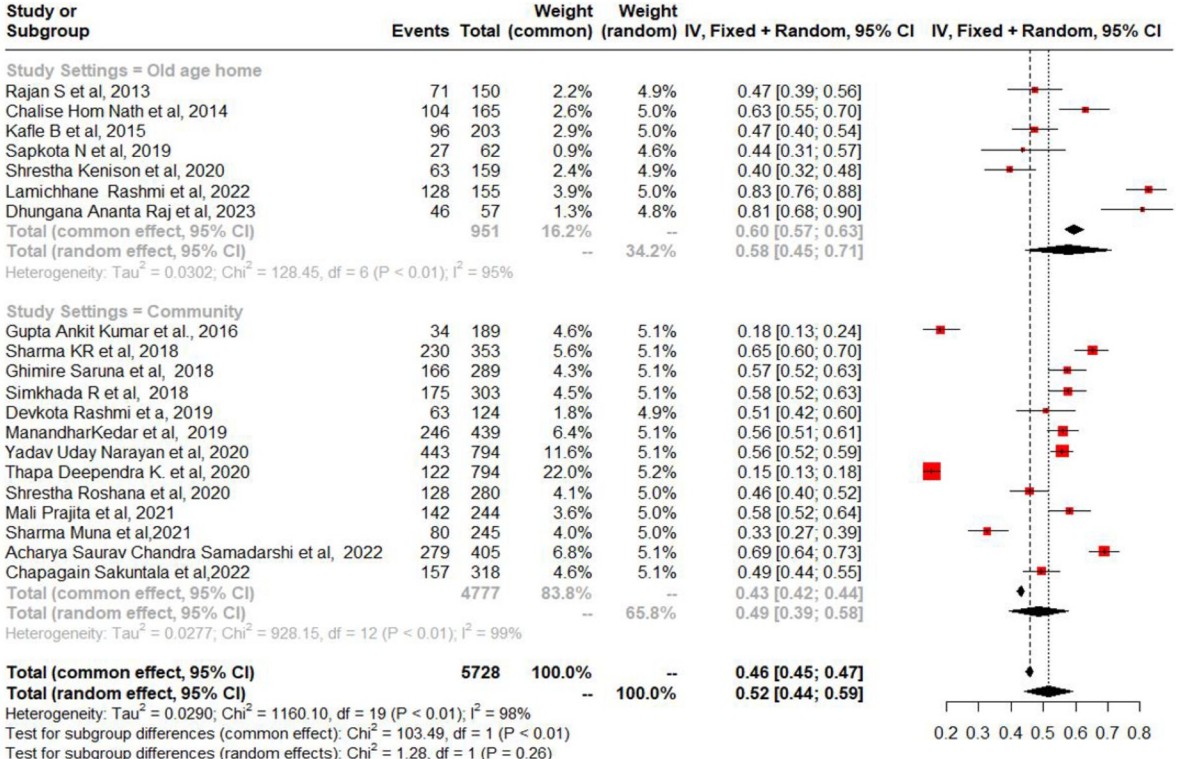

**Fig 4. Analysis of subgroups based on study settings.**

et al., 2017, 27.0% (95%, CI: 24, 29) [52] and Cai et al., 2023, 35.1% (95% CI: 30.2,40.4) [11] as well as the systematic review conducted in developing countries like India by Pilania et al., 2019, 34.4% (95%, CI:29.3, 39.7) [53] and Kasa et al., 2022, 41.85% (95%, CI: 33.52, 50.18) [33]. However, only one Iranian review article among elderly by Jafari & Ghasemi-Semeskandesh, in 2021 had reported results similar to our study, 52% (95%, CI: 46, 58) [54].

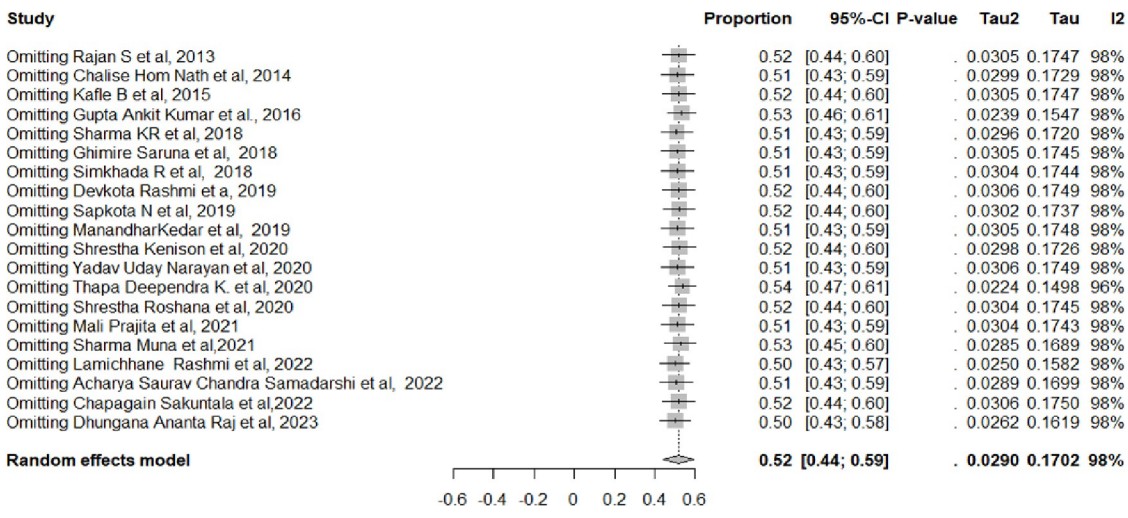

**Fig 5. Sensitivity analysis.**

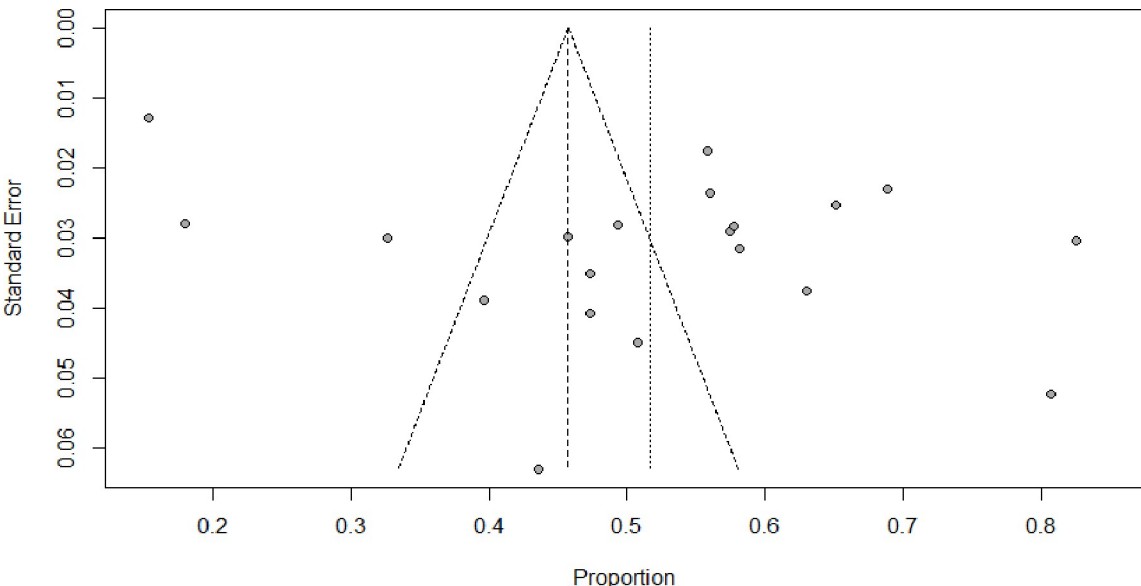

**Fig 6. Funnel plot.**

The prevalence of geriatric depression may vary depending on the study's settings, sample size, research methods, culture, values, and social norms, as well as the government's intended structures and the availability of elderly mental health services. The availability of mental health services, especially for the elderly, in some developed nations can reduce prevalence through early prevention and promotion [15, 55].

In this study, there is considerable regional differences in the prevalence of depression. Factors such as cultural contexts, levels of economic development, and geographic locations all had an impact on the occurrence of geriatric depression [56, 57]. Another explanation for this fact would be that all the 12 included studies from the central region had used GDS tool which obviously had a higher sensitivity [58]. The central development region of Nepal is a rapidly developing state compared to other regions and the government has established relatively modern effective mental health care levels which leads to a higher rate of identification of geriatric depression [18]. However, the health policies for geriatric care in Nepal is still at early stage [18, 59]. Similarly, we noticed that the prevalence of depression was higher in old age homes than in the community settings/centers/homes. Consistent results to our present study were observed in an Indian study by Mohan et al., 2015 [60] and a Nepalese study by Mali et al., 2021 [47]. The possible reason for this would be attributed to the fact that feeling of loneliness and being apart from family was commonly reflected in both studies [61].

Systematic reviews on the epidemiology of geriatric depression by Cai et al., 2023, Kasa et al., 2022 and Zenebe et al., 2021 [11, 15, 33] reinforced that geriatric depression is commonly envisaged by female gender, aged 75 and above, single, divorced or widow and with no educational background. The presence of chronic physical illness, limitation of IADL and lack of social support or respect, which was similarly reflected in our present study, further reinforced common determining factors ahead of geriatric depression. The results of the presents study are consistent with previous literature on socioeconomic status and health status with depression. A study conducted in six low and middle-income countries suggest that increasing age raised the chance of acquiring chronic diseases or multimorbidity, which is significantly associated with geriatric depression [62]. Furthermore, ageing also increased functional limitation which is positively associated with depression [62]. The most commonly used tool in order to

**Table 3. Factors associated with geriatric depression.**

| Factors associated | OR | 95% CI | Strength of association | Author, Year, of publication |
|---|---|---|---|---|
| Female sex | 1.37 | 1.01,2.10 | Strong | Chalise Hom Nath et al, 2014 [35] |
| Feeling of loneliness | 2 | 0.99,7.07 | Moderate | |
| Limitation in IADL | 1.67 | 1.21,2.3 | Strong | |
| Age ≥70 | 2.1 | 1.1,4.2 | Strong | Sharma KR et al, 2018 [38] |
| Female sex | 1.8 | 1.2,2.9 | Strong | |
| Unmarried/Widow | 1.9 | 1.1,3.1 | Strong | |
| Unsatisfied with the respect | 4.2 | 2.1,8.4 | Strong | |
| Illiterate | 2.01 | 1.08,3.75 | Strong | Simkhada R et al, 2018 [40] |
| Presence of chronic disease | 1.97 | 1.03,3.8 | Strong | |
| Limitation in IADL | 5.59 | 1.75,17.93 | Strong | |
| Age ≥75 | 1.9 | 1.3,2.9 | Strong | Manandhar Kedar et al, 2019 [43] |
| Illiterate | 3.1 | 1.7,5.1 | Strong | |
| Female sex | 1.3 | .90,1.9 | Strong | |
| Presence of chronic disease | 1.7 | 1.1,2.5 | Strong | |
| Single/window/divorced | 2 | 1.4,3.0 | Strong | |
| Perceived respect | 4.1 | 1.9,8.7 | Strong | |
| Presence of chronic disease | 1.24 | 1.05,1.46 | Strong | Thapa Deependra K. et al, 2020 [16] |
| Male sex | 0.51 | 0.29,.091 | Strong | |
| Smoking habit | 2.04 | 1.16,3.59 | Strong | |
| Perceived health status | 3.41 | 1.69,6.86 | Strong | |
| Disturb sleep | 2.3 | 1.28,4.24 | Strong | Shrestha Roshana et al, 2020 [46] |
| Experiencing pain | 1.3 | 1,1.74 | Strong | |
| Feeling of neglect/no respect | 2.4 | 1.09,5.29 | Strong | Mali Prajita et al, 2021 [47] |
| Age ≥75 | 3.04 | 1.21,7.57 | Strong | |
| Old age allowance | 2.35 | 1.03,5.37 | Strong | |
| Presence of chronic disease | 32.04 | 4.19,24.5 | Moderate | |
| Female sex | 2.01 | 1.16,3.38 | Strong | Sharma Muna et al,2021 [48] |
| Presence of chronic disease | 2.41 | 1.28,4.53 | Strong | |
| Age ≥70 | 2.26 | 1.42,3.57 | Strong | Acharya Saurav Chandra Samadarshi et al, 2022 [49] |
| Female sex | 2.76 | 1.78,4.30 | Strong | |
| Presence of chronic disease | 3.65 | 2.32,5.73 | Strong | |
| Low health service access | 4.72 | 2.75,8.11 | Moderate | |
| Age ≥70 | 1.43 | 0.889,2.28 | Strong | Chapagain Sakuntala et al,2022 [50] |
| Presence of chronic disease | 3.94 | 2.43,6.39 | Strong | |
| Single/window/divorced | 2.52 | 1.55,4.10 | Strong | |
| Illiterate | 4.33 | 2.62,7.13 | Strong | |
| Feeling of loneliness | 4.7 | 2.61,8.49 | Moderate | |
| Unsatisfied with the respect | 12.31 | 6.79,22.32 | Moderate | |

detect depression among included articles was Geriatric Depression Scale. Similar explanation was given by the number of other papers too [11, 15, 33, 52]. The possible reason for it may be due to its established reliability and validity, which can be used easily in community settings even in healthy or medically ill groups and is different from depression screening tools used in younger or adults groups [63].

## Strength and limitation

This meta-analysis is the first comprehensive review of geriatric depression among older adults in Nepal. This review was conducted following standard protocols and included studies using

a reliable diagnostic tool to measure geriatric depression which further assisted in generating unbiased results. Moreover, sensitivity analysis showed none of the included studies had an effect on overall findings. Nevertheless, there are some constraints in the present study. The significant heterogeneity represents the degree of variation among included studies. Most of the included studies (60%) were from the central region particularly from the Kathmandu district, therefore the derived result may be directive of that region. This concentration limits the generalizability of the findings to the entire country The present systematic review included only observational studies evidence to detect the covariates of geriatric depression which leads to contention with confounding. Hence, a more detailed analysis considering these confounders would strengthen the study's findings.

## Conclusion

The prevalence of geriatric depression was highly inflated with heterogenicity among older people of Nepal creating a soaring public health concern. Factors such as the presence of chronic diseases, females, illiteracy, senior age, lack of social support or respect and limitation of IADL were the independent predictors of geriatric depression among the older population. Therefore, it is essential to provide a high-priority plan for geriatric depression problems in order to generate more reliable national-wide data and formulate policies and plan in order to improve the mental health status of the growing older population in Nepal.

## Supporting information

**S1 File. PRISMA 2020 checklist.**
(DOCX)

**S2 File. Search strategy.**
(DOCX)

## Acknowledgments

We would like to acknowledge Health Action and Research Organization for providing necessary guideline for systematic search and risk of bias assessment.

## Author Contributions

**Conceptualization:** Gayatri Khanal, Y. Selvamani, Saravanan Chinnaiyan.

**Data curation:** Gayatri Khanal.

**Formal analysis:** Gayatri Khanal, Suman Thapa.

**Investigation:** Suman Thapa.

**Methodology:** Gayatri Khanal, Y. Selvamani, Suman Thapa, Saravanan Chinnaiyan.

**Software:** Gayatri Khanal, Saravanan Chinnaiyan.

**Supervision:** Gayatri Khanal, Y. Selvamani.

**Writing – original draft:** Gayatri Khanal, Suman Thapa, Saravanan Chinnaiyan.

**Writing – review & editing:** Gayatri Khanal, Y. Selvamani, Suman Thapa, Saravanan Chinnaiyan.

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
