## [Decision Letter · Decision Letter 0]

2 Jul 2024

PMEN-D-24-00071

Prevalence and Covariates of Depression among Older Adults in Nepal: A Systematic Review and Meta-Analysis

PLOS Mental Health

Dear Dr. Khanal,

Thank you for submitting your manuscript to PLOS Mental Health. After careful consideration, we feel that it has merit but does not fully meet PLOS Mental Health’s publication criteria as it currently stands. Therefore, we invite you to submit a revised version of the manuscript that addresses the points raised during the review process.

We look forward to receiving your revised manuscript.

Kind regards,

Mingrui Xia

Academic Editor

PLOS Mental Health

Journal Requirements:

1. If you did not receive any funding for this study, please simply state: “The authors received no specific funding for this work

Additional Editor Comments (if provided):

Reviewers' comments:

Reviewer's Responses to Questions

**Comments to the Author**

1. Does this manuscript meet PLOS Mental Health’s publication criteria? Is the manuscript technically sound, and do the data support the conclusions? The manuscript must describe methodologically and ethically rigorous research with conclusions that are appropriately drawn based on the data presented.

Reviewer #1: Yes

Reviewer #2: Yes

2. Has the statistical analysis been performed appropriately and rigorously?

Reviewer #1: Yes

Reviewer #2: Yes

3. Have the authors made all data underlying the findings in their manuscript fully available (please refer to the Data Availability Statement at the start of the manuscript PDF file)?

Reviewer #1: Yes

Reviewer #2: Yes

4. Is the manuscript presented in an intelligible fashion and written in standard English?

Reviewer #1: No

Reviewer #2: Yes

5. Review Comments to the Author

Reviewer #1: PLOS Mental Health

I have reviewed the manuscript entitled "Prevalence and Covariates of Depression among Older Adults in Nepal: A Systematic Review and Meta-Analysis." The manuscript addresses a significant public health issue in a country with limited studies. While the study highlights the prevalence and covariates of depression among older adults in Nepal, several improvements can enhance its comprehensiveness and clarity. Despite the high heterogeneity among the studies evaluated, which undermines the reliability and generalizability of the findings, the study is valuable for understanding geriatric depression in Nepal.

Introduction

Various information was included early in the text, but the order may be more understandable.

I recommend beginning the introduction with a concise and precise definition of depression. This should include its clinical characteristics, diagnostic criteria, and impact on daily life. Such a foundation will help readers understand the context of the study.

Highlight the specific ways in which depression manifests in older adults compared to younger individuals. Detail how symptoms may differ, why early detection is challenging, and the unique risk factors that older adults face. This differentiation is crucial for understanding why geriatric depression is a significant public health issue.

The authors may present statistical data on the prevalence of depression among older adults, particularly in developing countries. They should emphasize the lack of comprehensive data in such regions. This will underscore the importance of the current study and the need for updated epidemiological data.

Address the gaps in current research and the necessity for updated data on geriatric depression. Finally, explain the study's aim and how its findings can contribute to the literature.

Discussion

Discuss the geographic concentration of the studies included in the meta-analysis in detail. A significant proportion (60%) of these studies were conducted in the Central region of Nepal, particularly in the Kathmandu district. This concentration limits the generalizability of the findings to the entire country. It may not accurately reflect the prevalence of geriatric depression in other regions with different socio-economic and cultural contexts.

It is necessary to discuss the types of questionnaires most commonly used to detect depressive mood disorders in the cross-sectional studies included in the analysis. The detail will provide insight into the methodologies employed and their potential impact on the results.

Consider the potential confounding factors as a limitation of the study. While the paper identifies several risk factors for geriatric depression (e.g., female gender, chronic illness, lack of social support), it does not thoroughly address other factors that could influence these associations. For instance, socio-economic status, access to healthcare, and underlying health conditions might confound the relationships identified. A more detailed analysis considering these confounders would strengthen the study's conclusions.

Thank you for considering my feedback.

Sincerely,

Reviewer #2: The article is important because of its relevance and filling the gap in our knowledge in global mental health area. It provided the evidence to implement further public health interventions and send strong message on developing the better services for people with depression.

6. PLOS authors have the option to publish the peer review history of their article (what does this mean?). If published, this will include your full peer review and any attached files.

**Do you want your identity to be public for this peer review?** For information about this choice, including consent withdrawal, please see our Privacy Policy.

Reviewer #1: No

Reviewer #2: No

---

## [Decision Letter · Decision Letter 1]

1 Aug 2024

Prevalence and Covariates of Depression among Older Adults in Nepal: A Systematic Review and Meta-Analysis

PMEN-D-24-00071R1

Dear Mrs Khanal,

We are pleased to inform you that your manuscript 'Prevalence and Covariates of Depression among Older Adults in Nepal: A Systematic Review and Meta-Analysis' has been provisionally accepted for publication in PLOS Mental Health.

Best regards,

Mingrui Xia

Academic Editor

PLOS Mental Health

Reviewer Comments (if any, and for reference):

Reviewer's Responses to Questions

**Comments to the Author**

1. If the authors have adequately addressed your comments raised in a previous round of review and you feel that this manuscript is now acceptable for publication, you may indicate that here to bypass the “Comments to the Author” section, enter your conflict of interest statement in the “Confidential to Editor” section, and submit your "Accept" recommendation.

Reviewer #1: All comments have been addressed

2. Does this manuscript meet PLOS Mental Health’s publication criteria? Is the manuscript technically sound, and do the data support the conclusions? The manuscript must describe methodologically and ethically rigorous research with conclusions that are appropriately drawn based on the data presented.

Reviewer #1: Yes

3. Has the statistical analysis been performed appropriately and rigorously?

Reviewer #1: Yes

4. Have the authors made all data underlying the findings in their manuscript fully available (please refer to the Data Availability Statement at the start of the manuscript PDF file)?

Reviewer #1: Yes

5. Is the manuscript presented in an intelligible fashion and written in standard English?

Reviewer #1: Yes

6. Review Comments to the Author

Reviewer #1: I have had the opportunity to evaluate the study entitled "Prevalence and Covariates of Depression among Older Adults in Nepal: A Systematic Review and Meta-Analysis." The authors provide updated national estimates on geriatric depression among older adults in Nepal. The study's findings, revealing a significantly high prevalence rate of 52%, offer critical insights that could significantly inform and strengthen public health policies and mental health promotion.

Overall, the authors have effectively addressed my previous suggestions.

I recommend enhancing the introduction by including one or two sentences that discuss the differences in detection rates of geriatric depression between low—and middle-income countries. This addition would provide a broader context and emphasize Nepal's unique challenges.

The discussion section appropriately addresses the variation in prevalence rates across different regions and examines the impact of cultural, economic, and healthcare factors. This comprehensive approach provides a nuanced understanding of the factors influencing geriatric depression in Nepal.

7. PLOS authors have the option to publish the peer review history of their article (what does this mean?). If published, this will include your full peer review and any attached files.

**Do you want your identity to be public for this peer review?** For information about this choice, including consent withdrawal, please see our Privacy Policy.

Reviewer #1: No
